# Cardio-Oncology: Mechanisms, Drug Combinations, and Reverse Cardio-Oncology

**DOI:** 10.3390/ijms231810617

**Published:** 2022-09-13

**Authors:** Zehua Liang, Yuquan He, Xin Hu

**Affiliations:** China–Japan Union Hospital of Jilin University, Jilin University, Changchun 130033, China

**Keywords:** cardio-oncology, cardiovascular toxicity, cardioprotection, reverse cardio-oncology, traditional Chinese medicine

## Abstract

Chemotherapy, radiotherapy, targeted therapy, and immunotherapy have brought hope to cancer patients. With the prolongation of survival of cancer patients and increased clinical experience, cancer-therapy-induced cardiovascular toxicity has attracted attention. The adverse effects of cancer therapy that can lead to life-threatening or induce long-term morbidity require rational approaches to prevention and treatment, which requires deeper understanding of the molecular biology underpinning the disease. In addition to the drugs used widely for cardio-protection, traditional Chinese medicine (TCM) formulations are also efficacious and can be expected to achieve “personalized treatment” from multiple perspectives. Moreover, the increased prevalence of cancer in patients with cardiovascular disease has spurred the development of “reverse cardio-oncology”, which underscores the urgency of collaboration between cardiologists and oncologists. This review summarizes the mechanisms by which cancer therapy induces cardiovascular toxicity, the combination of antineoplastic and cardioprotective drugs, and recent advances in reverse cardio-oncology.

## 1. Introduction

As cancer mortality declines and the surviving population ages, the overlap between patients with heart disease and patients with cancer also increases [1]. Among cancer survivors, the risk of developing cardiovascular disease is higher than the risk of tumor recurrence, and the increased risk of cardiovascular disease is attributable to cancer treatment [2]. Cardiovascular-related complications such as heart failure (HF) and arrhythmia caused by anti-cancer drugs are relatively common. Among older survivors of breast cancer, the risk of dying from heart disease is greater than that from breast cancer itself [3]. Moreover, a cohort study of childhood cancer survivors showed that cardiotoxicity has become the second most important cause of long-term mortality after the tumor itself [4].

Daunorubicin was the first anthracycline (ANT) isolated from *Streptomyces peucetius*, followed by doxorubicin (DOX) isolation from *S. peucetius caesius* variant [5]. However, cardiotoxicity in DOX-treated pediatric patients was reported as early as 1976. Since then, several attempts have been made to separate the antitumor effects and cardiotoxic effects of ANTs, as well as to reduce toxicity further [6]. Simultaneously, the cardiotoxicity of ANT has drawn more attention. Studies have demonstrated that its toxic effects were dependent upon the cumulative dose, administration schedule, and age [7,8]. With in-depth understanding of the mechanism of cancer development, monoclonal antibodies, inhibitors, immunotherapy, and other therapies have emerged. However, while treating cancer efficaciously, the cardiotoxicity caused by these drugs has become a key problem affecting the survival, prognosis, and quality of life of patients. Hence, the field of “cardio-oncology” has advanced.

Reduced cardiac function from cancer therapy may be due to primary cardiomyopathy because of direct damage to myocardial cells. Secondary cardiomyopathy occurs due to changes in innervation or the hormonal milieu. Alternatively, myocarditis is caused by infiltration of inflammatory cells into the myocardium [9]. Cardio-oncology is based on the assessment and management of cardiotoxicity caused by anticancer therapy or the malignant process itself. The aim is to balance antitumor efficacy with cancer therapy-related cardiac events, thereby prolonging meaningful survival and providing supportive cardiovascular care for patients while striving for optimal cancer care [10]. For this purpose, researchers have focused on the prevention and reduction of the cardiotoxicity associated with cancer treatment. These efforts have included combining anti-cancer drugs and cardiovascular drugs without influencing the anti-cancer effect while achieving cardiac detoxification [11]: this is a new idea for the development of “tumor cardiology”. 

With the development of cardio-oncology, the close relationship between cancer and cardiovascular disease has been excavated continuously. The potential link between cardiovascular disease and subsequent malignancies is gaining attention [12]. Diseases such as hypertension [13], vascular embolism [14], and arrhythmia [15] have been associated with an increased risk of cancer, and myocardial infarction has been shown to promote progression of colorectal tumors [16]. Thus, “reverse cardio-oncology” is starting to gain traction.

Classical cardio-oncology focuses on the risk of cardiovascular disease in patients being treated for cancer. Reverse cardio-oncology is based the risk of subsequent cancer in patients with cardiovascular disease. Developments in these two directions suggest a complex crosstalk between the two diseases. Understanding this bidirectional relationship has important implications for the collaboration of cardiology and oncology as well as the prevention and treatment of related diseases. 

Here, we review the mechanisms of cardiotoxicity associated with cancer therapy, combination therapy, and some new developments in reverse cardio-oncology.

## 2. Cardiovascular Toxicity of Antitumor Therapy and Mechanisms

Treatments for advanced cancer are based on chemotherapy, radiotherapy, targeted therapy, and immunotherapy, all of which negatively affect the cardiovascular system and have been reported extensively [17]. In particular, chemotherapy- and radiotherapy-induced cardiotoxicities are the leading cause of morbidity and mortality in people surviving cancer [18]. In chemotherapy, there is well-documented cardiotoxicity associated with ANT use. In contrast, reports of cardiotoxicity associated with relatively novel therapies such as target therapy and immunomodulation are limited.

### 2.1. ANTs

ANTs are derived from *Streptomyces peucetius* var. caesius and include DOX, epirubicin, and daunorubicin [19]. ANTs were applied first in the clinic in the 1960s and have become one of the most widely used chemotherapy drugs. They continue to be used as first-line treatment of many solid cancers and hematological malignancies: acute leukemia, lymphoma, breast cancer, stomach cancer, and ovarian cancer [20,21,22].

#### 2.1.1. Antitumor Mechanism and Cardiotoxicity

The antitumor mechanism of ANTs is related to inducing DNA damage (directly or indirectly). They act mainly on proliferating cells in S and G2 phases. The parent nucleus of the anthracene ring is parallel to the base pair of DNA through non-specific insertion and forms a relatively stable complex. The parent nucleus is positively charged and has a high affinity with negatively charged DNA. The quinone structure in the molecule can participate readily in electron transfer reactions to generate oxygen free radicals [23]. On the one hand, ANTs are embedded in DNA and interfere with the replication and transcription of DNA. On the other hand, ANTs use leads to oxidative damage to nucleic acids and double-strand breaks of DNA by increasing the production of reactive oxygen species (ROS). In addition, ANTs interfere directly with the helicase activity and subsequent strand separation of DNA, as well as with topoisomerase type 2 (Top2) and DNA unwinding [24].

Cardiotoxicity is a dose-limiting side effect of ANTs, and the magnitude of toxicity increases with cumulative doses. For example, in a retrospective analysis, the risk of HF was increased significantly with cumulative doses of DOX up to 400 mg/m^2^; among patients receiving 500, 550, or 700 mg/m^2^ of DOX, the approximate prevalence levels of HF were 16%, 26%, and 48%, respectively [25]. Acute cardiotoxicity and chronic cardiotoxicity can be divided according to the time of onset. Acute cardiotoxicity induced by ANT therapy is rare and pathologically similar to that of acute toxic myocarditis with myocardial injury, interstitial edema, and infiltration of inflammatory cells [9]. In contrast, chronic cardiotoxicity caused by ANTs is more common, manifesting as dilated cardiomyopathy in experimental models and human hearts. The pathology is characterized by increased heart weight and dilatation of heart chambers. The most common pattern is the loss of intracellular myogenic fibers in human tissue and vacuolar degeneration with sarcoplasmic reticulum (SR) swelling and consolidation in animal models [26]. According to the time of onset, chronic cardiotoxicity is divided into early and late. Early-onset chronic cardiotoxicity, which develops within 1 year of treatment cessation, usually manifests as dilated and hypokinetic cardiomyopathy and leads to HF. Late-onset chronic cardiotoxicity develops years or even decades after chemotherapy ends [27]. The cardiotoxicity of ANTs involves oxidative stress, mitochondrial dysfunction, abnormal autophagy, and dysregulated homeostasis of calcium ions (Ca^2+^) (Figure 1).

#### 2.1.2. Imbalance in Redox Reactions

ROS have long been considered to be key mediators of ANT cardiotoxicity. ANTs can be reduced to activated hemiquinones by mitochondrial nicotinamide adenine dinucleotide (NADH) dehydrogenase, which produces superoxide anions in response to ROS exposure [28]. Due to the relative lack of oxygen-radical-scavenging enzymes (e.g., superoxide dismutase (SOD), catalase (CAT), or glutathione (GSH) peroxidase) in cardiomyocytes, upregulation of oxidative stress can be very deleterious to the heart [29]. If the redox reaction is unbalanced, heme in the heart is degraded, and free iron is released, accumulates in mitochondria, and triggers lipid peroxidation. Moreover, ANTs can form a complex with Fe^3+^ to catalyze the Fenton reaction. The latter generates H_2_O_2_, which can be converted into various ROS-related substances that eventually cause DNA damage response (DDR) and death of many cardiomyocytes [30,31]. Several studies have reported that the use of antioxidants, such as N-acetylcysteine, vitamin E, and coenzyme Q10 (CoQ10), can reduce the cardiotoxicity of ANTs. However, in some animal experiments, chronic use of antioxidants did not have the desired effect [32]. Therefore, the relative contribution of oxidative stress and production of primitive ROS to ANT-induced cardiotoxicity is incompletely understood.

#### 2.1.3. Disruption of Mitochondrial Function

Another interesting aspect related to ANT-induced cardiotoxicity is that ANTs show a particular predilection for mitochondria in cardiomyocytes [33]. Studies have shown that ROS production is related to mitochondrial dysfunction. DOX can bind directly to the abundant phospholipid cardiolipins on the inner mitochondrial membrane, inhibit complexes I and II, hinder the electron transport chain, and lead to ROS generation [34,35]. Then, ROS act on mitochondria, inducing dissipation of the mitochondrial membrane potential, causing mitochondria to swell, and prompting opening of the mitochondrial permeability transition pore (mPTP). Then, cytochrome c (Cyto c) is released from mitochondria to activate the apoptotic pathway and induce mitochondria-dependent cardiomyocyte apoptosis [35,36,37]. Thus, a “vicious circle” is formed. ROS destroys the electron transport chain in mitochondria, leading to further ROS formation that, in turn, acts on mitochondria, aggravating mitochondrial dysfunction and Cyto c release, and aggravating the intrinsic apoptotic pathway in cardiomyocytes. Tadokoro et al. [37,38] demonstrated that DOX increases lipid peroxides (LPs) in the mitochondria of cardiomyocytes but not in other organelles. The inhibition of LPs in mitochondria reduces DOX-induced ferroptosis. This phenomenon occurs because DOX downregulates glutathione peroxidase 4 (GPx4) expression in mitochondria. Under normal conditions, GPx4 inhibits LPs by chelating DOX–Fe^2+^ complexes, so DOX causes GPx4 downregulation and LP excess, which induces ferroptosis. DOX-induced mitochondrial toxicity can also be due to dysregulation of the autophagic response, but whether the specific cause is autophagy inhibition or excessive autophagy is not clear. 

On the one hand, DOX-mediated inhibition of autophagy leads to the accumulation of dysfunctional mitochondria and subsequent death of cardiomyocytes [39]. In healthy condition, the stress response induces PTEN-induced kinase 1 (PINK1) aggregation in the outer mitochondrial membrane, and the translocation of Parkin and p62 to mitochondria triggers mitophagy to digest damaged mitochondria. In the presence of DOX, abnormal mitochondrial structure and mitochondrial damage have been shown to occur in neonatal rat cardiomyocytes. DOX disrupts mitophagy by inhibiting the interaction of Sestrin2 (SESN2) with Parkin and p62 [40]. Moreover, DOX-induced inhibition of mitochondrial autophagy may be related to cytoplasmic p53, which binds to Parkin and inhibits its translocation to mitochondria to inhibit autophagy. Mice lacking P53 show relatively weak inhibition of mitochondrial autophagy after DOX intervention [41]. 

On the other hand, some studies have suggested that cardiotoxicity is due to excessive autophagy (including mitophagy) induced by DOX [42]. Yin et al. demonstrated that DOX activates mitophagy. It is manifested by mitophagosomes, an increased microtubule-associated protein 1 light chain 3 (LC3) II/I ratio and beclin-1 levels, decreased p62 levels, and activation of the PINK1/Parkin pathway, which promotes PINK1/Parkin translocation to mitochondria. In addition, inhibition of mitophagy attenuates mitochondrial dysfunction and protects cardiomyocytes from death [43]. In addition to mitophagy, the mechanism by which DOX induces cardiotoxicity by affecting autophagy, “macroautophagy” (hereafter referred to as autophagy), is discussed later. 

#### 2.1.4. Autophagy Abnormality

The role of autophagy in cardiotoxicity induced by antitumor therapy has been controversial. Autophagy is a lysosome-dependent bulk degradation mechanism and is essential for maintaining cellular homeostasis [44]. Autophagy starts by activating the adenosine 5‘-monophosphate -activated protein kinase (AMPK) pathway and inhibiting the mammalian target of rapamycin (mTOR) signaling pathway [45]. By regulating these two pathways, the activity of unc-51-like autophagy activating kinase 1 (Ulk-1) is precisely controlled. Ulk-1 can phosphorylate beclin-1 that, in turn, promotes the formation of autophagy-related complexes; triggers autophagosome formation; captures substances (e.g., proteins and organelles) that need to be cleared; and, finally, fuses with lysosomes to degrade or recycle damaged components [46,47,48]. Early reports demonstrated that ANT treatment increased autophagy, thereby mediating adverse cardiac events. The specific mechanism is known. Briefly, ANTs activate AMPK, upregulate the downstream autophagy marker beclin-1 [49], promote the formation of autophagic vacuoles, and increase the LC3-II/I ratio [50]. Consistently, the hearts of beclin-1^+/−^ mice (pure and lethal) were shown to be protected from ANT-induced damage, which suggested that inhibition of autophagy initiation was cardioprotective [51]. Indirectly, inhibition of the anti-apoptotic protein B-cell lymphoma-2 (Bcl-2), a negative regulator of beclin-1, leads to autophagy and cell death. Upregulation of Bcl-2 expression by the transcription factor GATA4 (GATA-binding protein 3) can suppress autophagy-related gene expression and attenuate DOX-induced cardiotoxicity [52]. 

In contrast, other reports have emphasized ANT-induced cardiotoxicity due to inhibition of autophagy. Li et al. [53] found that DOX activates the phosphoinositide 3-kinase gamma/protein kinase B (PI3Kγ/Akt) signaling pathway downstream of Toll-like receptor-9 (TLR9), leading to activation of mTOR that, in turn, inhibits Ulk-1 and inhibits autophagy initiation. Consistent with this hypothesis, Li et al. [54] found that DOX treatment induced insufficient activation of cardiac autophagy. This action caused accumulation of LC3-II, p62, and ubiquitinated proteins, and impaired myocardial autophagic flux, ultimately causing cardiotoxicity. In addition, some evidence for a link between autophagy and ANT cardiotoxicity points to lysosomes. Autophagy is a dynamic process that requires a comprehensive assessment rather than a “snapshot in time”. DOX can affect not only autophagy initiation and autophagosome maturation but can also influence the subsequent fusion of lysosomes and exercise of their functions. Li et al. [51] demonstrated, for the first time, that DOX interferes with lysosome function by inhibiting luminal acidity. This action results in the accumulation of defective autolysosomes, thereby inhibiting autophagic flux in cardiomyocytes. On the basis of this premise, Bartlett et al. [55] demonstrated that DOX inhibits macroautophagy and lysosomal proteolysis by inhibiting EB (a transcription factor that controls lysosomal signaling and function), resulting in the injury and death of cardiomyocytes. The role of autophagy in DOX-induced cardiotoxicity is difficult to precisely define, and an accurate analysis of autophagic flux is lacking in those studies. Increased LC3-II levels may reflect enhanced autophagosome formation or hindered fusion between autophagosomes and lysosomes [56]. 

The cardiotoxicity of DOX is dose-related, and there are some differences in the clinical manifestations of acute cardiotoxicity and chronic cardiotoxicity induced by DOX. The lack of studies on dose and modeling duration in studies may also be the reason for controversial results. However, regardless of whether DOX induces cardiotoxicity by activating or inhibiting autophagy, the reversal of autophagy at that time can reduce cell death [57], which is very interesting.

#### 2.1.5. Targeting Top2 

Top2 is required for the replication and transcription of DNA. Top2 catalyzes the unwinding of the supercoiled DNA double helix, thereby controlling the topological state of DNA [58]. The role of Top2 is to cleave one strand of duplex DNA, allowing the passage of a second duplex strand, reannealed and as an intermediate in this process. Covalent binding occurs between DNA and Top2 to form a cleavable complex [59]. If DOX is present, DOX inserts into the “minor groove” of duplex DNA alongside Top2 and stabilizes the cleavable complex [60]. Stabilization of the cleavable complex leads to stalling of the enzymatic activity of Top2 and induction of DNA double-strand breaks, leading ultimately to apoptosis [61,62]. 

There are two isoenzymes of Top2 in humans: Top2α and Top2β [63]. Top2α shows high expression in rapidly proliferating cells and peaks during the G2/M phase of the cell cycle. Top2β is expressed in cardiomyocytes and other quiescent normal cells. Therefore, inhibition of Top2 not only leads to DNA double-strand breaks in rapidly proliferating cells but also induces DNA damage in cardiomyocytes [33,64]. Lyu et al. [65] isolated primary mouse embryonic fibroblasts from Top2β^−/−^ mice with DOX intervention and demonstrated that they were resistant to DOX-induced cell death. Zhang et al. [33] found that DOX-induced DDR may induce upregulation of *p53*-related genes and downregulation of peroxisome-proliferator-activated receptor gamma coactivator 1 (*Pgc1*), leading to defective mitochondrial biogenesis and metabolic abnormalities. However, cardiomyocyte-specific deletion of Top2β protects mice from these effects of DOX, indicating that DOX-induced cardiotoxicity is mediated by Top2β in cardiomyocytes. Furthermore, DOX was able to cause a 50% downregulation in expression of PGC-1α and PGC-1β in the cardiomyocytes of wild-type mice, whereas PGC-1 transcripts were unchanged in Top2β^−/−^ cardiomyocytes [66]. Notably, the transcriptional coactivators PGC-1α and PGC-1β are vital regulators of mitochondrial biogenesis in models of HF [67,68,69]. In addition, PGC-1α is one of the critical regulators of SOD expression [70]. Therefore, DOX may cause the inhibition of PGC-1α expression by targeting Top2β that, in turn, increases ROS production by inhibiting SOD expression. Under the toxicity of low concentrations of DOX, Ras-related C3 botulinum toxin substrate 1 (Rac1) has also been shown to regulate how DOX targets Top2. Moreover, inhibition of the Rac1 signaling pathway prevents the formation of DNA–Top2 cleavable complexes, thereby reducing the genotoxic potency of DOX targeting Top2, reducing subsequent DSB formation and activation of the DDR [71]. 

#### 2.1.6. Dysregulation of Ca^2+^ Homeostasis

Dysregulation of Ca^2+^ homeostasis is a pathological event in ANT-based cardiotoxicity. In cardiomyocytes, Ca^2+^-dependent signaling is highly regulated and determines the strength of myocardial contraction, whereas regulation of the intracellular Ca^2+^ concentration ([Ca^2+^]_i_) in the heart is disturbed during DOX treatment [72]. An early report found that mRNA expression of Ca^2+^ transporters in the SR and plasma membrane was reduced significantly in DOX-treated rabbit hearts, as well as reduced Ca^2+^-ATPase protein and reduced Ca^2+^ uptake in the SR, which resulted in impaired Ca^2+^ handling [73]. Ca-ATPase of the SR is mainly responsible for regulating the [Ca^2+^]_i_ of cardiomyocytes during excitation–contraction, so the systolic and diastolic functions of the heart are affected [74]. In addition, DOX exposure in vivo increases total Ca^2+^ content in the SR, leading to Ca^2+^ overload [75]. DOX can also induce an overload of intracellular Ca^2+^ channels by increasing Ca^2+^ influx through L-type calcium channels [76]. In a study of the metabolic determinants of ANT-induced cardiotoxicity, DOX *per se* was shown to downregulate cardiac-specific transcriptional regulators, resulting in reduced expression of ryanodin receptor-2 (RyR2), α-actin, and slow myosin light chain 2 [77,78]. 

What is more, metabolites of DOX affect Ca^2+^ dysregulation significantly. DOX is converted in vivo to DOXOL by two-electron carbonyl reductase. DOXOL cannot diffuse readily from cardiomyocytes to plasma due to its high polarity and can accumulate in the heart. Moreover, its inactivation of ATP-dependent Ca^2+^ handling proteins and inhibition of *RyR2* is 30–40 times higher than that of DOX [79]. Moreover, DOXOL inhibits sodium–calcium exchange channels, resulting in myocardial-energy imbalance and weakened systolic function [80]. Notably, DOX-induced Ca^2+^ dysregulation is closely related to oxidative stress. Calcium/calmodulin-dependent protein kinase II (CaMKII) can be activated through oxidation, and Ca^2+^ leakage from the SR causes myocardial dysfunction. Moreover, isolated cardiomyocytes perfused with DOX show increased occurrence of Ca^2+^ “sparks”, decreased Ca^2+^ transients, and decreased cell contractility [81], which further demonstrates the damaging effect of DOX on regulation of the Ca^2+^ concentration.

In addition to cytosolic Ca^2+^ dysregulation, DOX also causes changes in mitochondrial Ca^2+^ flux, which leads to mitochondrial dysfunction. DOX treatment significantly reduces mitochondrial Ca^2+^-loading capacity in rat cardiomyocytes [82]. Ca^2+^ enters mitochondria through the mitochondrial Ca^2+^ uniporter. If Ca^2+^ exceeds the physiological threshold, mitochondrial Ca^2+^ triggers opening of the mPTP. Then, the mitochondrial membrane depolarizes, resulting in increased Ca^2+^ influx into mitochondria. This action causes mitochondrial damage and increased permeability of the outer membrane to apoptotic factors such as Cyto c, which activate caspase-3 and lead ultimately to apoptosis [24,83]. Blockade of the mitochondrial Ca^2+^ cycle protects cardiomyocytes from Ca^2+^ intolerance. Hence, altered regulation of mitochondrial Ca^2+^ transport is also a key event in DOX-induced cardiomyopathy [84].

#### 2.1.7. Changes in Epigenetic Modification

Epigenetic modification mainly regulates the function and expression of genes through DNA methylation, histone modification, regulation of non-coding RNA (ncRNA), and chromosome remodeling [85]. The critical role of epigenetic mechanisms in complex diseases (including cardiovascular disease) is well established [86]. The cardiotoxicity caused by ANTs may be a long-term disease course. Hence, in recent years, some researchers have focused on epigenetic changes, finding that the abnormal epigenetic modification caused by DOX treatment may be related to its cardiotoxicity.

Hanf et al. [87] found that in DOX-treated cardiomyocytes, expression of histone lysine demethylases (Lysine-specific histone demethylase 3A (KDM3A) and lysine-specific demethylase 1 (LSD1)), histone lysine methyltransferases (histone-lysine N-methyltransferase 7 (SET7) and histone-lysine N-methyltransferase (SMYD1)), and histone deacetylases (sirtuin 1 (SIRT1) and histone deacetylase 2 (HDAC2)) were altered significantly and affected post-translational modifications. Consistent with this hypothesis, Ferreira et al. [88] demonstrated that DOX treatment reduced global DNA methylation in the heart. These differences were accompanied by alterations in mRNA expression of multiple gene groups. In addition, expression of DNA methyltransferase 1 is downregulated under oxidative stress, and DNA methylation of the mitochondrial genome is maintained by DNA methyltransferase 1, which may lead ultimately to demethylation of the mitochondrial genome [89]. There is also evidence that histone modifications are associated with DOX-induced cardiotoxicity. An experiment in human cardiomyocytes (AC16 cells) showed that DOX induced PGC-1α acetylation and repressed expression of the genes related to the function and biogenesis of mitochondria, such as mitochondrial transcription factor A, SOD2, cytochrome c oxidation enzyme IV, and nuclear respiratory factor-1 (NRF-1) [90]. Furthermore, DOX treatment resulted in upregulation of HDAC6 expression, resulting in deacetylation of α-tubulin, and inhibition of HDAC6 expression exhibited cardioprotective effects [91].

Abnormal changes in ncRNA expression have also been observed after DOX intervention. For example, DOX treatment upregulates miR-208a and leads to downregulation of GATA4, resulting in increased apoptosis and decreased cardiac function. Silencing miR-208a expression rescues GATA4, reduces apoptosis, and improves cardiac function [92]. In addition, downregulation of CircITCH (circRNA generated from several exons of itchy E3 ubiquitin protein ligase) expression has been found in ANT-treated, human-derived pluripotent stem cell-derived cardiomyocytes as well as in autopsy specimens from patients with cancer-induced cardiomyopathy. Moreover, CircITCH overexpression alleviated DOX-induced injury and dysfunction of cardiomyocytes, and this relief was dependent on its inhibitory effect on miR-330-5p [93]. In addition, overexpression of miR-212/132 clusters has been shown to prevent cardiotoxicity in DOX-induced mouse models [94].

In conclusion, DOX-induced cardiotoxicity has been shown to be accompanied by abnormal epigenetic alterations in vitro and in vivo. Although the causal relationship between epigenetics and ANT-induced cardiotoxicity has not been clearly established, reversing abnormal epigenetics could protect the heart from the toxic effects of ANTs.

### 2.2. Non-ANT-Based Chemotherapy

In addition to ANTs, other types of chemotherapy drugs can also cause cardiovascular toxicity: alkylating agents, platinum drugs, and antimetabolites [95] (Figure 2).

#### 2.2.1. Platinum

Platinum drugs such as *cis*-diamminedichloroplatinum(II) (cisplatin) have been used to treat testicular, bladder, lung, ovarian, and other types of cancer [96]. Cisplatin is activated after entering cells. In the cytoplasm, cisplatin hydrolyzes and produces electrophiles that react with nitrogen donor atoms on nucleic acids. This action causes DNA damage in cancer cells, blocks cell division, and leads to cell death. These hydrolyzed forms inhibit mitochondrial respiration by uncoupling oxidative phosphorylation, resulting in Ca^2+^ outflow and disruption of cellular homeostasis [97]. In addition, cisplatin generates ROS by inducing oxidative stress, and the main target is also the mitochondrion [98]. 

The pathways by which cisplatin activates apoptosis are complex and diverse. They include activation of the p38-mitogen-activated protein kinase (MAPK) and Janus kinase (JNK) pathways, regulation of extracellular-signal-regulated kinase (ERK) and PI3K/Akt signaling cascades, as well as other mechanisms [97]. On the basis of these mechanisms, while cisplatin kills cancer cells, it also kills cardiomyocytes. On the one hand, it may be directly toxic to cardiomyocytes. On the other hand, excessive ROS generation induces oxidative stress in cardiomyocytes. Consistent with this hypothesis, cardiovascular events such as arrhythmias, cardiomyopathy, myocarditis, and thromboembolism caused by cisplatin therapy have been reported [99,100]. El-Awady et al. [101] found that cisplatin can reduce the activity of cardiac SOD and the level of GSH and increase the level of malondialdehyde (MDA). These events cause oxidative stress reactions, which leads to leakage of cardiac enzymes and cardiac troponin I, and increases indices of serum cardiotoxic enzymes, such as lactate dehydrogenase (LDH) and creatine kinase (CK). 

Similar to ANTs, cisplatin also causes mitochondrial dysfunction, endoplasmic reticulum (ER) stress, and apoptosis [102]. However, the common side effects of cisplatin in clinical practice are not HF but are reflected more deeply in the vascular system [103]. Following cisplatin treatment, platinum remains in the blood to form “circulating platinum”. Studies have shown that in patients with testicular cancer treated with cisplatin, circulating platinum may continue to act on the endothelium to cause endothelial damage and increase levels of fibrinogen, tissue plasminogen activator, high sensitivity C-reactive protein (hs-CRP), plasminogen activator inhibitor, and von Willebrand factor to increase the risk of atherosclerosis [104,105]. In addition, cardiomyocytes produce various proinflammatory factors and chemokines after cisplatin administration, leading to neutrophil infiltration, nuclear factor-kappa B (NF-κB) translocation, and increased tumor necrosis factor-alpha (TNF-α) production [106], which may lead ultimately to cardiomyocyte/tissue inflammation.

#### 2.2.2. Alkylating Agents

The discovery of “nitrogen mustard” as an alkylating agent in the 1940s ushered in a new era of chemotherapy [107]. Alkylating agents are cytotoxic drugs that can form compounds with active electrophilic groups in the body and then inactivate them by combining with abundant electronic groups in DNA and RNA. Moreover, alkylating agents can crosslink DNA double chains, prevent replication, and eventually cause cell death [108]. 

Commonly used alkylating agents such as cyclophosphamide (CP) have dose-related cardiotoxicity, which often causes side effects such as atrioventricular block, tachyarrhythmia, myocarditis, or HF if taken in high (100–200 mg/kg bodyweight) doses [109]. The lowest dose of CP that showed cardiotoxicity was 46.5 mg/kg. A patient with diffuse large B-cell lymphoma exposed to this dose died of congestive HF. The sudden HF of this patient may have been the result of accumulation of several ANTs in the early stage of treatment [110]. Doses of 120–200 mg/kg (i.v.) are considered cardiotoxic according to clinical studies using CP [111]. 

CP is a physiologically inactive drug that requires a complex metabolic process to function, which also becomes a barrier to predicting and treating its side effects. The exact mechanism by which CP causes cardiotoxicity is incompletely understood [112]. The main pathway of CP metabolism is hydroxylation to form 4-hydroxycyclophosphamide, which can decompose to form acrolein and phosphoramide mustard [109]. Recent studies have shown that human cardiomyocytes exhibit cytotoxicity if exposed to CP at concentrations 10 times higher than those administered clinically [107]. Concentration of CP causes cardiotoxicity is much higher than that in clinical use. Thus, the cardiotoxicity of CP might be attributable to its metabolites. 

CP has been shown to be metabolically activated in rats. It was shown to significantly reduce the levels of cardiac antioxidant enzymes SOD and catalase; increase serum CK, LDH, and MDA; lead to downregulation of *Nrf2* (NF-E2-related factor 2) and upregulation of *Keap1* (Kelch-like ECH-associated protein 1); and inhibit the PI3K/Akt/mTOR signaling pathway. CP induces cardiomyocyte injury and oxidative stress. Moreover, histopathology studies have revealed degeneration of myocardial tissue, vacuolization of myocardial cells, and infiltration of inflammatory cells [113,114]. As one of the products of CP metabolism, acrolein induces nitrative stress in the myocardium. The phosphorylation of endothelial nitric oxide synthase (eNOS) at Ser^1177^ was shown to be inhibited in a long-term intervention of acrolein in mouse hearts; eNOS was uncoupled, the number of dimers was reduced significantly, and monomer levels were increased to induce cardiac nitrative stress [115]. Furthermore, acrolein can be transferred to the heart through the circulation to produce protein–acrolein adducts. The latter cause protein modifications (cardiac α-actin, desmin, and myosin light polypeptide 3), elicit significant effects on the mitochondrial energy metabolism of contractile/cytoskeletal proteins, and disrupt myofilament function selectively [116].

#### 2.2.3. Antimetabolites

In the 1940s, studies on folate antagonists for the remission of childhood leukemia heralded the era of antimetabolites. In the 1950s, 6-mercaptopurine (which inhibits adenine metabolism for the treatment of acute leukemia) and 5-fluorouracil (5-FU; a uracil analog for non-hematological cancers) were studied [117]. 

Antimetabolites are analogs of natural metabolites that have a role in cellular metabolism. In general, they are compounds similar to the normal building blocks of DNA and RNA. Folates are also involved in the synthesis of DNA and RNA, and folate analogs are also considered to be antimetabolites [118]. If an antimetabolite diffuses into a tumor cell, enzymes in the purine or pyrimidine metabolic pathway are replaced with cellular nucleotide analogs. The latter inhibit DNA-synthesis-related enzymes, cause DNA damage, and induce apoptosis [119]. Antimetabolites have also been associated with cardiotoxicity [120].

Methotrexate (MTX) was the first antimetabolite to be approved by the U.S. Food and Drug Administration. MTX has shown contradictory cardiac effects. MTX has been shown to reduce the risk of cardiovascular disease by altering nucleotide metabolism and attenuating (at least in part) cytokine signaling, as well as reducing chronic inflammation [121,122]. Low-dose MTX has been shown to reduce levels of TNF-α and interleukin-6 (IL-6) and increase the level of IL-10 in the plasma of rats with cardiac myosin-induced autoimmune myocarditis, as well as increase left-ventricular ejection fraction and fractional shortening [123]. Ridker and colleagues undertook a randomized, double-blind trial of low-dose MTX or matching placebo in 4786 patients with prior myocardial infarction or multivessel coronary artery disease who additionally had type-2 diabetes mellitus or metabolic syndrome. MTX did not reduce levels of IL-1 β, IL-6, or CRP, and did not result in fewer cardiovascular events [124]. 

The cardiovascular-protective effects of MTX have shown mixed results, but additional studies have demonstrated the cardiotoxicity of MTX. MTX-treated zebrafish embryos exhibited cardiac and vascular dysplasia, as well as reduced heart rate and ventricular fractional shortening. These affects may be related to the reduced transcript levels of the genes with central roles in cardiac morphogenesis, cardiac development, ventricular contractility, and angiogenesis [125]. In rats, MTX was shown to induce oxidative damage to tissue by reducing the activity of antioxidant enzymes in cardiac tissue and increasing lipid peroxidation. Moreover, compared with a control group, MTX caused a significant reduction in expression of Bcl-2 protein and a significant increase in expression of p53 and cluster of differentiation (CD)68, indicating that cardiomyocytes appeared to undergo apoptosis [126]. In addition, the cardiotoxicity of MTX may be related to the inflammatory response. The possible mechanism is that MTX stimulates TLR4 expression, which activates the NF-κB pathway and induces the production of proinflammatory cytokines such as TNF-α and IL-6 [127].

5-FU was the earliest anti-pyrimidine drug. Capecitabine was developed as a prodrug of FU, which is metabolized to 5-FU to function [128]. The cardiac side-effects of 5-FU affect 0% to 30% of patients. The most commonly reported symptom is chest pain, but arrhythmias and myocardial infarction may also occur [129]. 5-FU can be metabolized to 5-fluorodeoxyuridine acid, which blunts DNA synthesis by inhibiting thymidylate synthase and can also be transformed into fluorouridine 5′-triphosphate to inhibit RNA synthesis [130]. 5-FU is catabolized mainly by dihydropyrimidine dehydrogenase, and its products include α-fluoro-β-alanine (FBAL) and fluoroacetic acid. One report revealed that FBAL might be associated with the cardiotoxicity of 5-FU. A patient with cancer without a history of cardiac disease presented with precordial pain with right bundle branch block during intravenous infusion of 5-FU. A sharp increase in the serum level of FBAL to 1955 ng/mL was noted; pain disappeared after discontinuation of 5-FU, and electrocardiography changes disappeared. After switching to S-1 (derivative of 5-FU that can inhibit dihydropyrimidine dehydrogenase, which catalyzes the degradation of 5-FU), the serum concentration of FBAL decreased to 352 ng/mL, and subsequent cardiac symptoms were not observed [131]. In addition, the relatively non-toxic fluoroacetate may be converted to fluorocitrate, which inhibits aconitase and succinic dehydrogenase of the tricarboxylic acid (TCA) cycle [132]. Analysis of intermediates of the TCA cycle revealed accumulation of citrate within the 5-FU-treated myocardium, suggesting that dysfunction in the TCA cycle caused by fluorocitrate inhibition of aconitase is responsible for high-energy phosphate depletion [133]. Consistently, after 5-FU administration in rats, its product acts on mitochondria, blocks the TCA cycle, reduces ATP production, and alters the permeability of the mitochondrial membrane, which causes mitochondrial dysfunction. Furthermore, 5-FU upregulates expression of PINK-1, Parkin, and beclin-1 and downregulates expression of Bcl-2, causing excessive mitochondrial autophagy and leading to myocardial injury [134]. 

In addition, the cardiotoxicity caused by 5-FU may be related to oxidative stress. Cardiac tissue and cardiomyocytes exposed to 5-FU show reduced activity of antioxidant enzymes, accumulation of lipid-peroxidation products, and an impaired antioxidant defense system [128,135]. With respect to the direct cardiovascular toxicity of 5-FU, some studies have found that 5-FU causes echinocytosis and increases the fluidity of the erythrocyte membrane, which reduces blood viscosity. Moreover, 5-FU causes K^+^ efflux to result in arrhythmia, as well as rapid depletion of O_2_ and increased levels of 2,3-bisphosphoglycerate, which result in decreased levels of ATP, cardiac ischemia, and hypoxia [136]. 

Furthermore, studies have attributed the mechanism of 5-FU-induced cardiotoxicity to direct toxic effects on the vascular endothelium. These include affecting eNOS (leading to coronary spasms), through Akt (to induce endothelium-independent vasoconstriction), and increasing endothelin-1 levels to increase coronary vascular resistance [137,138]. Polk et al. [139] proposed a mechanism of 5-FU-induced cardiotoxicity. Briefly, thrombosis after endothelial injury leads to increased metabolism. This action causes energy expenditure and ischemia, oxidative stress (leading to cell damage), and coronary-artery spasm (leading to myocardial ischemia and a reduced ability of red blood cells to transfer oxygen). Other clinical antimetabolites for tumor treatment have also been reported to be associated with cardiotoxicity, such as HF caused by fludarabine and pericarditis caused by cytarabine. However, research on their mechanism of action is lacking [140,141,142].

### 2.3. Radiotherapy

Radiotherapy and chemotherapy, along with surgery, are the key modalities used in cancer treatment. Radiotherapy is an efficacious treatment for cancers of the breast, thyroid gland, prostate gland, head, and neck [143,144,145]. Radiotherapy causes DNA damage mainly by ionizing molecules in irradiated tissue, blocking DNA replication, and killing tumor cells [146]. Each Gray (1 Gy) of radiation absorbed by each cell produces 105 ionizations, which results in≈∼2000 single-strand breaks and 40 DSBs in DNA, in addition to other types of DNA damage, such as damage to nuclear nucleotide bases, DNA–DNA crosslinking, and DNA–protein crosslinking [146]. 

The relationship between chemotherapy and heart damage was first linked in 1983. The radiation-related cardiovascular complications that have been identified include valvular heart disease, arrhythmias, right-ventricular damage, HF, pericardial disease, and coronary artery disease [147]. Radiation-induced cardiovascular toxicity is dose related. The development of cardiovascular disease and inflammation is accelerated if the radiation dose reaches 1–4 Gy. Radiation doses of 5–8 Gy increase the prevalence of angina, myocardial infarction, and pericarditis, whereas radiation doses >8 Gy can cause fibrosis of cardiac muscles [148]. After cardiomyocytes have been irradiated, the stimulated ER releases Ca^2+^ from the Ca^2+^ pool of the ER into the cytoplasm, which causes mitochondrial Ca^2+^ overload [149]. Moreover, radiation-induced oxidative stress can induce the production of endogenous ROS in mitochondria and activate p53 simultaneously, causing genome instability and permanent dysfunction in mitochondria [150,151]. Excessive ROS and p53-dependent upregulation of expression of PUMA (also known as bbc3 (BCL2 binding component 3)) and (Bak (BCL2 antagonist/killer)/Bax (BCL2 associated X)) lead to increased permeabilization of the mitochondrial membrane, followed by the release of Cyto c into the cytoplasmic matrix, which activates the apoptosome complex and triggers apoptosis. 

Mitochondria are the main organelles for energy supply in cardiomyocytes [150,151,152]. If mitochondria are dysfunctional, cardiomyocyte activity will decrease significantly. If cells are under chronic oxidative stress for a long time, the number of adhesion molecules and proinflammatory mediators increases [153]. Studies have shown that exposure of the heart to radiation leads to increased deposition and release of von Willebrand factor in endothelial cells and increased levels of p-selectin, e-selectin, platelet-endothelial cell adhesion molecule-1, and intercellular cell adhesion molecule-1. Expression of proinflammatory adhesion factors promotes acute inflammation, increases the probability of thrombosis, and leads to myocardial ischemia and tissue inflammation [149,154]. In addition, radiotherapy can cause capillary rupture, leading to microvascular damage, thickening of blood vessel walls, and atherosclerosis [155].

In animal models, long-term irradiation can induce increased thickening of the left ventricular wall, decreased ventricular diameter, and increased levels of extracellular-matrix-associated transforming growth factor beta (TGF-β) [156]. TGF-β/SMAD2/3 (suppressor of mothers against decapentaplegic 2/3) are considered key molecular mechanisms of radiation-induced cardiac fibrosis [157]. Long-term irradiation has been shown to cause overexpression of the fibrotic markers connective tissue growth factor, TGF-βRII, and SMAD2/3, and increased collagen content in the left ventricle, thereby indicating fibrosis. In addition, long-term irradiation can induce significant increases in ratios of pERK1/tERK1, pERK2/tERK2, and pAKT/tAKT, thereby suggesting compensatory cardiac hypertrophy mediated by non-canonical SMAD-independent signaling pathways [158]. Moreover, high-dose radiotherapy can lead to pericarditis, possibly because of the accumulation of damage-associated molecular patterns in the pericardium due to radiation. Damage-associated molecular patterns induce TLR expression, thereby stimulating NF-κB, which subsequently stimulates activation of the inflammasome, leading to the release of proinflammatory cytokines. Moreover, NF-κB can induce expression of cyclo-oxygenase-2, which in turn promotes ROS release [159] (Figure 3).

### 2.4. Targeted Therapy

The era of targeted therapy began with the approval of trastuzumab, a monoclonal antibody against human epidermal growth factor receptor 2 (HER2), for the treatment of HER2-positive breast cancer [160]. The emergence of targeted therapy solves (at least in part) the problem of damage to healthy cells caused by the non-specific pharmacological effects of traditional chemotherapeutic drugs. However, the cardiotoxicity of targeted therapy has emerged gradually with an increase in clinical use (Figure 4). Clinically targeted cancer drugs are divided mainly into two categories: monoclonal antibodies and small molecules [161].

Monoclonal antibodies bind to the specific antigens of cancer cells or block the binding of ligand receptors and inhibit the growth of cancer cells [161]. Trastuzumab (also known as erbB2) is a humanized monoclonal antibody directed against the extracellular domain of HER2. Trastuzumab has been shown to improve the prognosis of women with HER2-positive breast cancer by preventing the activation of intracellular tyrosine kinases by binding to the extracellular portion of the HER2 receptor [162]. However, along with its antitumor effect, cardiotoxic side effects are also common. Farolfi et al. [163] reported the prevalence of cardiac events associated with trastuzumab treatment to be as high as 44%. In a meta-analysis of trastuzumab-treated HER2-positive breast cancer, trastuzumab-treated patients were 2.45 times more likely to have cardiotoxicity compared with patients not taking adjuvant trastuzumab [164]. 

Although its mechanism of cardiotoxicity is complex and elusive, several studies have provided insights. Trastuzumab blocks erbB2, which plays an important part in cardiomyocyte survival. Blockade of erbB2 results in increased ROS levels in cardiomyocytes and mPTP-dependent (mitochondrial Cyto c is released) cell death. In addition, Akt phosphorylation is inhibited in erbB2–antibody-treated cardiomyocytes, suggesting that the cardiotoxic effect may be related to inhibition of the AKT pathway [165]. Elzarrad and colleagues showed that trastuzumab-treated mice developed myocardial-fiber rupture and reduced numbers of myocardial mitochondria, which resulted in cardiac damage. Moreover, trastuzumab can cause myocardial oxidative stress and activate an apoptotic pathway by increasing the activity of caspase3/7 [166]. 

Bevacizumab is a humanized monoclonal antibody against vascular endothelial growth factor A (VEGF-A). It was the first targeted angiogenesis inhibitor to be developed [167]. Bevacizumab use has been reported to increase the prevalence of hypertension [168], arrhythmia [169], and other diseases. In vitro, bevacizumab-induced cardiomyocyte injury manifests as decreased viability of cardiomyocytes; increased [Ca^2+^]_i_; increased cellular mitochondrial ROS levels; mitochondrial swelling; collapse of the mitochondrial membrane; attenuated energy metabolism; and activation of ER stress. Cardiomyocyte death may be caused by activation of apoptotic pathways due to increased expression of Bax, Bad, and caspase-9; endogenous apoptosis caused by mPTP opening; and mitochondrial damage caused by inhibition of the ERK pathway and oxidative stress [170,171].

Most small-molecule drugs have been developed as tyrosine kinase inhibitors (TKIs). The latter can penetrate cell membranes, interfere with signaling pathways, and act on intracellular targets [161]. Cardiotoxicity-related drugs such as osimertinib (target is the epidermal growth factor receptor EGFR), gefitinib (EGFR), sunitinib (multi-target), and sorafenib (multi-target) have been reported. 

Among TKIs targeting the EGFR, osimertinib carries a relatively higher risk of cardiotoxicity, but the underlying mechanism of action is not known. In a retrospective analysis by Anand et al. [172] comparing the risk of cardiotoxicity between osimertinib and other drugs in the Adverse Events Reporting System of the U.S. Food and Drug Administration, 6.1% of adverse events with osimertinib were cardiac related. Osimertinib was associated with an increased risk of HF, QT prolongation, atrial fibrillation, and pericardial effusion compared with all other drugs. In addition, Oyakawa et al. [173] reported a case of osimertinib-induced myocarditis resulting in irreversible heart damage. Similarly, gefitinib, a competitive inhibitor of activation of receptor tyrosine kinase targeting the EGFR (ErbB1), has relatively weak toxicity but, because of its early clinical use, the reports on cardiac events caused by it are relatively detailed [174]. Lynch Jr et al. [175] reported a case of recurrent myocardial infarction with angiographically documented rupture of vulnerable plaques in a patient receiving long-term gefitinib therapy for a metastatic carcinoid tumor. In addition, in a cohort study of 27,992 patients based on treatment with tyrosine-kinase-targeted drugs and/or chemotherapy, gefitinib use was shown to increase the HF risk, which suggested that gefitinib use may increase the risk of adverse cardiovascular events [176]. In vitro, gefitinib led to increased phosphatase and tensin homolog (PTEN) expression in cardiomyocytes, and synergistic inhibition of EGFR activity resulted in blockade of the PI3K/AkT pathway, which led to increased expression of the downstream forkhead box O3a (FoxO3a) gene to induce cardiotoxicity [177]. In a long-term intervention study using gefitinib in mice, cardiac damage developed. This event resulted in significant increases in levels of creatine kinase-MB (CK-MB), the N-terminal part of brain natriuretic peptide (NT-proBNP), and cardiac troponin I. A response to oxidative stress was also observed by activation of JNK and p38 pathways and reduction in expression of antioxidant enzymes, as well as inhibition of phosphorylation of ERK1/2 and Akt to blunt intracellular signaling [178]. The examples given above are small-molecule-targeted drugs for the erbB receptor family. TKIs targeting the erbB receptor family also include lapatinib (erbB1/2), neratinib (erbB1/2/4), afatinib (erbB), and dacomitinib (erbB1/2/4). They have been reported to induce cardiotoxicity at a relatively low overall level [179,180,181,182]. 

In addition to the examples described above, many multitargeted TKIs are also associated with cardiotoxicity. Sunitinib is a small-molecule TKI that can inhibit receptor tyrosine kinase family members containing split kinase domains, including VEGFR 1/2/3, platelet-derived growth factor receptors A and B, and the stem cell factor receptor [183]. In a review of 75 patients with a metastatic gastrointestinal stromal tumor treated with sunitinib, cardiovascular events occurred in 11% and congestive HF in 8% of cases [184]. Furthermore, Richards et al. assessed 6935 sunitinib-treated patients and found that the overall prevalence levels for all patients and patients with high-grade congestive HF were 4.1% and 1.5%, respectively, which suggested that sunitinib use is associated with an increased risk of congestive HF in cancer patients [185]. The cardiovascular toxicity of sunitinib is hypothesized to be the result of off-target inhibition of receptor tyrosine kinase and mitochondrial function [186]. Sunitinib-induced cardiomyocyte injury manifests as myocardial mitochondrial swelling, cardiomyocyte apoptosis, oxidative stress, and inflammation development. Sunitinib induces impaired function of the respiratory chain, reduces the mitochondrial membrane potential, stimulates ROS production, and causes cells to undergo oxidative stress, which lead to increased caspase3/7 activity and apoptosis promotion. Exposure to sunitinib results in significant increases in the gene expression of IL-1b, IL-6, and NF-κB, which leads to activation of inflammatory pathways [187,188]. 

In addition, some studies have used the concept of autophagy to explain the toxicity of sunitinib to cardiomyocytes, but the role of autophagy in cardiotoxicity is controversial. Zhao et al. [189] found that H9c2 cells treated with sunitinib for 48 h died, but apoptosis markers in cells were consistently negative, whereas autophagy markers were increased significantly, and cell death was reduced significantly after knockdown of beclin-1 expression to block autophagy. They speculated that cardiomyocyte autophagy may be one of the mechanisms of sunitinib cardiotoxicity. However, in other reports, the cause of sunitinib-induced cardiotoxicity was shown to be due to inhibition of autophagy. Yang et al. [190,191] demonstrated that sunitinib inhibited cardiomyocyte autophagy by inhibiting AMPK activation and promoting mTOR activation and induced the death of cardiac pericytes by inhibiting the GSTP1 (glutathione S-transferase pi 1)/JNK/autophagy pathway. This controversy may have arisen due to the different doses and duration of sunitinib action, or the different analytical criteria for autophagic flux. 

When exploring the cardiotoxicity of TKIs, sorafenib merits attention. Abdel-Rahman et al. found that patients receiving sorafenib had a significantly increased risk of hypertension and bleeding [192]. Early studies demonstrated that inhibition of the Raf/MEK (MAP kinse-ERK kinase)/ERK (extracellular regulated MAP kinase) pathway by sorafenib was the key to promoting apoptosis in zebrafish cardiomyocytes, and that oxidative stress was one of the reasons for cardiotoxicity induction [193,194]. Recent studies have shown that sorafenib can induce ventricular arrhythmia and left-ventricular dysfunction in isolated rat hearts and induce the disruption of Ca^2+^ homeostasis and myocardial damage by inhibiting mitochondrial complex III and mPTP opening, as well as causing overactivation of calcium/calmodulin-dependent protein kinase II [195].

### 2.5. Immunotherapy

Immunotherapy is designed to enhance natural defenses to eliminate malignant cells. It was developed on the basis of research into “tumor escape” mechanisms, and its emergence has revolutionized oncology and is a major breakthrough in cancer treatment. In addition, recent research has combined immunotherapy with mathematical models to optimize treatment for patients on the basis of their tumor progression [196]. Immunotherapy mainly includes immune checkpoint inhibitors (ICIs), adoptive cell therapy, immune-cell-targeted therapy using monoclonal antibodies, and others [197,198]. ICIs are antibodies that block negative regulators of T-cell immune responses. The most widely used ICIs are anti-cytotoxic T lymphocyte antigen-4 (CTLA-4), anti-programmed death protein-1 (PD-1), and anti-programmed death-ligand 1 (PD-L1) antibodies [199,200]. 

PD-1 is a co-inhibitory molecule expressed on T cells, B cells, monocytes, and activated natural killer cells. If PD-1 binds to its ligand PD-L1, the proliferation and migration of T cells are inhibited [201]. Often, PD-L1 is expressed in tumors or in the tumor microenvironment. After binding to PD-1 molecules on the surface of infiltrating T lymphocytes at tumor sites, PD-L1 inhibits T-cell activity and realizes immune evasion of tumor cells [202]. Therefore, PD-1- or PD-L1-targeted antibody therapy can reactivate the immune response at the tumor site and kill the tumor. 

CTLA-4 is a co-inhibitory molecule expressed on regulatory T cells as well as conventional T cells. CTLA-4 competes with CD28 to bind B7 (including CD80 and CD86), leading to the cellular immune response being “turned off”, and CTLA-4 blockade causes the expansion and activation of T cell clones [201,203]. ICIs can cause immune-mediated adverse events, among which events associated with cardiovascular toxicity include pericardopathy, vasculitis, arrhythmias, HF, myocarditis, and myocarditis. Immune-related adverse cardiovascular events studied using VigiBase (Global Individual-Case-Safety-Report database of the World Health Organization) suggested that the mortality rate of myocarditis caused by ICIs was up to 50%, and that patients receiving ICIs carried a risk of myocarditis 11 times higher than that in patients not receiving ICIs [201,204]. The specific mechanism of cardiotoxicity induced by cancer immunotherapy is incompletely understood, but its toxicity mechanism is traceable (Figure 5).

PD-L1-deficient autoimmune mice were shown to have infiltration of more inflammatory cells in the heart and more deaths from congestive HF than controls, and these mice developed autoimmune myocarditis spontaneously similar to that observed in humans [205]. Consistently, in a PD-1-deficient model of autoimmune disease in mice, a certain percentage of mice died of myocarditis [206]. Moreover, anti-PD-1-treated patients developed myocarditis with rhabdomyolysis, and T cells infiltrating the heart were identical to T-cell clones in the tumor, suggesting that nonspecific T-cell autoimmunity mediated myocarditis [207]. Like PD-1/PD-L1, CTLA-4 is important for maintaining the balance of self-tolerance and immune response in various tissues. T cells in CTLA-4-deficient mice were activated spontaneously and proliferated, and developed lymphoproliferative disorders rapidly, which mediated tissue damage with severe myocarditis [208]. Likewise, CTLA-4 ablation may lead to the proliferation and infiltration of CD8^+^ T cells in the heart, resulting in severe myocarditis [209].

Other immunotherapies, such as chimeric antigen receptor (CAR) T-cell therapy, have been reported to be associated with sinus tachycardia, new-onset arrhythmias, and decompensated HF. However, whether they are due to the direct cardiotoxic effect of CAR T cells has not been defined, presumably because of the widespread immune and inflammatory activation pathways [210,211]. Activation of CAR T cells results in massive systemic release of cytokines, including IL-6, interferon-γ, and TNF-α. This event leads to prostaglandin activation and triggering of cytokine release syndrome. Moreover, increased levels of IL-6 drive the inflammatory response in cytokine release syndrome, which may be one of the mechanisms leading to cardiovascular toxicity [212]. In addition, reports of cardiotoxicity from other immunotherapies, such as bispecific T-cell engager therapy, are infrequent or unclear [9].

## 3. Combination of Drugs to Achieve Cardiac Detoxification

Previous studies focused on using combination therapy to enhance the killing effect on cancer cells, such as increasing the sensitivity of oral squamous cell carcinoma to radiotherapy and the poly (ADP-ribose) polymerase inhibitors by pharmacological depletion of the function of chromatin assembly factor-1 complex [213]. In addition, drugs that target chromosomal rearrangements in tumors can also be used in combination therapy to customize treatment or improve efficacy [214]. However, due to the side effects of drugs, including cardiotoxicity, cancer patients must stop taking the drugs or change their treatment plans constantly, which increases the difficulty of combination therapy [215]. The occurrence of cardiovascular disease has become a major problem affecting treatment and prognosis of cancer patients. Along the same lines, many groups tried to reduce the cardiac burden of antitumor therapy and maintain the antitumor effect of the treatment by combining drugs. Some cardiovascular protective drugs are administered clinically to reduce the effects of cardiotoxicity. This part of our review focuses on the protective effects of TCM formulations and natural medicines on the cardiovascular toxicity caused by cancer treatment. 

Due to the different levels of tolerance and physical status of patients, the same treatment plan has different effects on different patients [216]. In contrast, TCM emphasizes enhancing the body’s resistance to disease, especially in individualized and combined treatment [217]. Moreover, different ingredients in TCM formulations can achieve cardioprotective effects through different mechanisms. Many TCM formulations are compatible, so “personalized treatment” may be expected.

### 3.1. Common Clinical Drugs

Various guidelines on clinical practice have been published to address the cardiotoxicity caused by anticancer therapy. Their contents mainly include a summary of cardiovascular complications of anticancer drugs; methods to assess cardiotoxicity; assessment of cardiovascular risk/prevention in cancer patients; screening and monitoring of cardiac function during cancer treatment; management of pre-existing cardiac disease to facilitate the most efficacious cancer treatment; and active management of cardiotoxicity due to anticancer therapy [218,219,220]. Often, cardioprotective agents are administered concomitantly to patients when managing the cardiotoxicity caused by cancer therapy. Here, we summarize some of the agents in clinical use or being employed in preclinical studies.

Dexrazoxane is a derivative of ethylenediaminetetraacetic acid. It protects the heart by chelating with intracellular iron, which can reduce ANT-induced production of free radicals [221]. In vitro, the scavenging capacity of dexrazoxane for hydroxyl radicals has been reported to be 320% higher than that of GSH and 100% higher than that of Trolox [222]. The cardioprotective effects of dexrazoxane have been studied clinically for more than two decades, and it is one of the most widely studied cardioprotective agents [223]. Cellular and animal studies suggest that dexrazoxane increases cell viability and protects cardiac function by reducing the apoptosis and necroptosis of cardiomyocytes following DOX treatment and attenuates DOX-induced inflammation and necroptosis by inhibiting the p38MAPK/NF-κB pathway [224]. Dexrazoxane-treated patients have been shown to have a significantly lower cumulative probability of DOX-induced cardiotoxicity, significantly fewer cardiac events, and a lower prevalence and severity of congestive HF than a control group [25,225,226]. 

Spironolactone is an aldosterone antagonist that blocks the final step of the renin–angiotensin–aldosterone system. Aldosterone antagonists have a positive effect on cardiac fibrosis and are often used in severe HF and myocardial infarction [227,228]. Animal studies have shown that spironolactone exhibits cardioprotective effects by attenuating DOX-induced myocardial fibrosis, apoptosis, and myocardial collagen volume fraction [229]. Spironolactone has antioxidant effects and protects the myocardium from ANT-related myocardial injury in breast-cancer patients [230]. Spironolactone can maintain mitochondrial ultrastructure and reduce ROS production [231]. 

Carvedilol is a β-receptor blocker. Animal studies have shown that carvedilol prevented ANT-induced cardiomyopathy and hydroxyl radical-induced cardiac contractile dysfunction and avoided ANT-induced apoptosis [232,233,234]. The inhibition of oxidative stress, mitochondrial dysfunction, and histopathological damage elicited by carvedilol is due to its intrinsic antioxidant activity [235]. A prospective, randomized, double-blind, placebo-controlled study evaluating carvedilol in the prevention of ANT-induced cardiotoxicity showed no effect on the prevalence of early-onset reduction in left-ventricular ejection fraction. However, its use resulted in a significant reduction in troponin levels and diastolic dysfunction [236].

Enalapril is an angiotensin-converting enzyme inhibitor (ACEI). ACEIs can slow the progression of left-ventricular systolic dysfunction. In addition, ACEIs play an important part in regulating Ca^2+^ homeostasis and improving the positive inotropic response of the myocardium to β-adrenergic stimulation [237]. Addition of enalapril to the routine treatment of patients with severe congestive HF has been shown to reduce mortality and improve symptoms [238]. Animal studies have shown that ACEIs can improve serum levels of glutamate oxaloacetate transaminase (GOT), glutamic-pyruvic transaminase (GPT), CK-MB, and LDH significantly in DOX-treated rats; those results suggest that enalapril has antioxidant potential to avoid DOX-induced cardiotoxicity [239]. In a randomized, single-blind, placebo-controlled study in which 69 patients newly diagnosed with malignancy and scheduled for ANTs were assigned randomly to receive enalapril or placebo, left-ventricular end-systolic volume and left atrial diameter were increased significantly compared with baseline levels in the control group, and enalapril appeared to help maintain systolic and diastolic function in cancer patients treated with ANTs [240].

CoQ10 (also known as panthenol) is synthesized endogenously in the smooth ER via the mevalonate pathway. It is an antioxidant whose primary role is as a component of energy production in the mitochondrial respiratory chain [241]. As an antioxidant, CoQ10 is believed to reduce the severe oxidative stress caused by semiquinones (ANT metabolites) in the inner mitochondrial membrane [242]. Animal experiments have shown that CoQ10 can achieve cardioprotective effects by preserving mitochondrial function during reperfusion [243]. CoQ10 has also been shown to alleviate significant reductions in cardiac weight and lipid peroxidation, restore cardiomyocyte alignment and increased collagen levels, and improve fibrosis and cell-death-related proteins to attenuate DOX-induced changes in cardiac tissue [244,245]. A study in children showed that Q10 reduced left-ventricular shortening percentage and ventricular septal thickening, thereby demonstrating the protective effect of CoQ10 on cardiac function during DOX treatment. That is direct evidence of the cardiac-protective ability of CoQ10 in preclinical studies [246]. Those studies demonstrated the potential of Q10 as an adjuvant to anti-cancer treatment.

### 3.2. Natural Products/TCM Formulations

Scholars have investigated the protective effects of natural drugs against cardiotoxicity induced by anticancer treatment. There are about 422,000 species of flowering plants worldwide, of which >50,000 have medicinal value and are used for pharmaceutical purposes. Approximately 80% of the global population relies on traditional medicines to meet primary healthcare needs [247]. Natural medicines are crucial preventive/therapeutic approaches, but their efficacy has not been demonstrated in systematic preclinical studies. The results of recent studies, which have focused mainly on animals and cell lines, demonstrate their potential for future application (Table 1). 

#### 3.2.1. Phenolic Acids

Phenolic acids owe their antioxidant activity mainly to their phenolic hydroxyl group [248]. They can also enhance endogenous antioxidant defense, activate the transcription factor Nrf2 via JNK-mediated phosphorylation, and then induce anti-oxidant reactive element (ARE)-dependent antioxidant/detoxification phase-II enzymes such as glutathione reductase (GR) and GPx [249]. Phenolic acids from plants have been shown to have cancer-protective effects by scavenging harmful free radicals and stimulating carcinogen excretion through the activation of phase I and II enzyme systems involved in the elimination of compounds from the body [250]. In addition to relying on antioxidant pathways, phenolic acids can also inhibit cancer by stimulating apoptosis and activating the mitochondrial pathway [251], inhibiting angiogenesis, and preventing migration of tumor cells [252,253]. The immunomodulatory and vasodilatory properties of polyphenols may also contribute to a reduced risk of cardiovascular disease [254], thereby reflecting their cardioprotective effects. Hence, phenolic acids may have a protective effect against chemotherapy-induced cardiotoxicity.

Salvianolic acid is the main active ingredient of *Salvia miltiorrhiza*, which is used in TCM formulations. In mice with DOX-induced acute heart injury, salvianolic acid was found to increase bodyweight and heart weight/tibial length ratio, reduce CK levels, and improve electrocardiography and cardiac-vacuole formation. Those effects are thought to be related to the antioxidant capacity of salvianolic acid, as reflected by its ability to reduce MDA levels [255]. He et al. [37] found that curcumin activates Bad (S112) phosphorylation by upregulating 14-3-3γ expression. This event led to Bcl-2 translocation into mitochondria, blockade of mPTP opening, and ROS-induced ROS release inhibited excessive oxidative stress and reduced DOX-induced cardiomyocyte death. Resveratrol (RSV) protects rat cardiomyocytes from DOX-induced cardiotoxicity by upregulating expression of the VEGF-B/Akt/GSK-3β (glycogen synthase kinase-3 beta) signaling pathway, thereby inhibiting cardiomyocyte apoptosis [256]. RSV can reduce DOX-induced cardiotoxicity by regulating autophagy, and RSV has certain flexibility in the regulation of autophagy. If DOX induces autophagy in cardiomyocytes, RSV reduces cardiotoxicity by inhibiting ribosomal protein S6 kinase 1 (S6K1) expression, thereby inhibiting autophagy [257]. If DOX reduces autophagy in cardiomyocytes, RSV can restore the impaired autophagy function by inhibiting mTOR or activating AMPK and SIRT1 pathways [258].

#### 3.2.2. Polysaccharides

Polysaccharides are biopolymers composed of monosaccharides linked together by glycosidic bonds. These structures can be linear or contain branched side chains [259]. As biological macromolecules, polysaccharides can mediate antitumor effects by stimulating T cells or other types of immune cells [260]. Herbal polysaccharides can protect the heart from injury in myocardial ischemia–reperfusion injury by increasing endogenous antioxidants, inhibiting cardiomyocyte apoptosis, modulating the inflammatory response, increasing energy production, and reducing infarct size [261]. The cardioprotective and anticancer capacity of polysaccharides predicts their value in combination with other anticancer drugs.

Xu et al. [262] demonstrated that pretreatment with *Ganoderma lucidum* polysaccharides inhibited Cul3 (cullin 3)-mediated K48 (Lysine 48)-linked polyubiquitination of Nrf2 by inhibiting Clu3 expression. These actions stabilized *Nrf2* expression, resulting in modification of MDM2 (murine double minute2), P53, and heme oxygenase-1 (HO-1), as well as inhibition of the NF-κB signaling pathway, and inhibited DOX-induced death, apoptosis, oxidative stress, and proinflammatory cytokine production in cardiomyocytes. Cao et al. showed that *Astragalus* polysaccharide could activate the PI3K/Akt signaling pathway and inhibit the p38MAPK pathway. Moreover, *Astragalus* polysaccharide inhibited expression of proapoptotic proteins such as caspase 3 and caspase 9 in vitro and in vivo, and reduced DOX-induced cardiomyocyte apoptosis [263]. Abushouk et al. demonstrated that *Ganoderma atrum* polysaccharide could increase the activity of manganese superoxide dismutase, reduce caspase activity, and reduce cardiomyocyte death. In addition, *Ganoderma atrum* polysaccharide reduced the release of Cyto c from mitochondria to the cytoplasm and attenuated DOX-induced mitochondrial damage by increasing the mitochondrial membrane potential and inhibiting mPTP opening [264].

#### 3.2.3. Flavonoids

The basic structure of flavonoids consists of an aromatic A ring and heterocyclic C ring, which are linked to the aromatic B ring by a C-C bridge [265]. A review of clinical trials has shown that flavonoids can be used to treat various types of tumors [266]. Flavonoids exhibit their anticancer activity by inhibiting protein kinases; inhibiting pro-oxidant enzymes; regulating the metabolism of carcinogens; inhibiting drug resistance; anti-angiogenesis; and inducing apoptosis and cell cycle arrest [267]. Moreover, flavonoid-rich foods and beverages can prevent cardiovascular disease by improving endothelial function, increasing the bioavailability of nitric oxide and lowering blood pressure [268]. Pretreatment with plant flavonoids can prevent ischemia–reperfusion injury [269]. Those studies have demonstrated the cardioprotective effect of flavonoids. A combination of flavonoid and classical anticancer drugs have shown improved efficacy in cancer treatment and also exhibited cardioprotective effects. Long-term efforts may continue to explore the effects and mechanisms of action of flavonoids alone or in combination [270].

Mantawy et al. [271] demonstrated that chrysin prevents DOX-induced oxidative stress by restoring cellular defense mechanisms, inhibiting DOX-induced mitochondrial-dependent apoptotic cell death by blunting p53, p38, JNK, and NF-κB signaling pathways, and that the VEGF/PI3K/Akt signaling pathway is enhanced to promote cardiomyocyte survival. Qi et al. [272] showed that cardamonin improves Nrf2-dependent antioxidant signaling by activating Nrf2 activity and inhibiting Nrf2 degradation from inhibiting oxidative stress and protects the heart from DOX-induced cardiotoxicity by reducing DOX-induced cardiomyocyte apoptosis and inflammation by reducing expression of caspase-3 and NF-κB signaling pathways, respectively. Zhai et al. [273] found that calycosin increased the viability of H9c2 cells and reduced DOX-induced apoptosis by regulating Bcl-2, Bax, and PI3K/Akt signaling pathways. In addition, calycosin improved levels of Sirt1–NOD-like receptor protein 3 and related proteins in cells and mice hearts, thereby reducing cardiotoxicity.

#### 3.2.4. Saponins

Saponins are steroidal or triterpenoid glycosides characterized by their soap-forming properties. Several studies have reported the role of saponins in cancer and their mechanisms, including cell cycle arrest, antioxidant activity, cell invasion inhibition, apoptosis, and induction of autophagy [274]. Saponins have also been used as adjuvants in cancer vaccines [275,276]. Animal experiments have suggested saponins can protect against myocardial ischemia–reperfusion injury [277,278]. Predictions based on network pharmacology have suggested that saponins can promote mRNA expression of VEGF-A, thereby inhibiting early apoptosis of vascular endothelial cells in coronary artery disease [279]. On the basis of the anticancer and cardioprotective effects of saponins, several scholars have evaluated their attenuating effects on cardiotoxicity induced by antitumor therapy in mice.

Wang et al. [280] demonstrated that the ginsenoside Rg3 could improve DOX-induced decline in ejection fraction and fractional shortening, as well as improve left-ventricular outflow. Rg3 can promote cell viability by activating Akt to activate the Nrf2/ARE pathway to alleviate DOX-induced oxidative damage and apoptosis. Lin et al. [281] demonstrated that astragaloside IV attenuated DOX-induced ventricular fibrosis and left-ventricular systolic dysfunction and significantly reduced cardiomyocyte death, apoptosis, and myocardial cell death by inhibiting nicotinamide adenine dinucleotide phosphate oxidase 2 (NOX2)- and NOX4-mediated oxidative stress. Saikosaponin D is an anticancer agent derived from the Chinese herb *Radix bupleuri*. Saikosaponin D has been shown to reduce DOX-induced LDH leakage, cardiomyocyte apoptosis, myocardial fibrosis, and reduction in cardiomyocyte size in mice. Saikosaponin D can increase levels of endogenous antioxidant enzymes, reduce levels of MDA and ROS to inhibit excessive oxidative stress, and protect cardiomyocytes from DOX-induced cardiotoxicity by inhibiting the p38-MAPK signaling pathway [282].

#### 3.2.5. Alkaloids

Alkaloids have a wide range of biological activities. In general, they are extracted from the plants of the families Leguminosae, Ranunculaceae, Papaveraceae, Solanaceae, Oleaceae, Rutaceae, and Polygonaceae, and they are also present in some animals [283,284,285]. Alkaloids can exert anti-tumor properties by inhibiting the proliferation, migration, and invasion of cells; triggering cell cycle arrest; reducing cytoskeletal stability; inhibiting epithelial–mesenchymal transition; and regulating various genes (e.g., upregulation of expression of tumor suppressors and interacting with various pathways) [286]. Alkaloids also have inhibitory effects on chronic inflammation, an essential player in the tumor process [287]. Some alkaloids also exhibit a protective effect against myocardial reperfusion deficit injury [288], isoprenaline-induced cardiotoxicity [289], and protection against propafenone-hydrochloride-induced acute HF [290]. Berberine is a candidate against cancer [291] and cardiovascular disease [292], demonstrating the potential of alkaloids as anti-cancer-therapy-induced cardiotoxic agents [293].

Priya et al. [294] stated that neferine can inhibit mitochondrial autophagy by activating insulin-like growth factor 1 receptor (IGF-1R) signaling and its downstream PI3K/Akt/mTOR pathway. Neferine can also induce Nrf2 translocation and increase expression of SOD-1 and HO-1 to suppress oxidative stress, thereby ameliorating DOX-induced cardiotoxicity. Wu et al. [295] showed that berberine can improve DOX-induced ST, QRS complexes, and QT-interval prolongation. Berberine can improve the activity of antioxidant enzymes and upregulate SIRT1 expression and downregulate expression of adaptor protein p66shc to inhibit ROS production. They suggested that the SIRT1/p66shc pathway might be a new target for reducing DOX toxicity. Cyclovirobuxine D is a triterpenoid alkaloid derived from *Buxus microphylla*. It can ameliorate DOX-induced oxidative damage in the heart, including lipid peroxidation and protein carbonylation, and reduce the ratio of reduced GSH to oxidized glutathione. It also prevents DOX-induced disorders in mitochondrial biogenesis.

#### 3.2.6. TCM Formulations

TCM theory has been used in clinical practice for over 3000 years. With regard to cancer treatment, TCM formulations can relieve symptoms (e.g., fatigue, chronic pain, anorexia/malady, and insomnia); improve quality of life; and reduce the adverse effects and complications caused by chemotherapy, radiotherapy, and targeted therapies [296]. In recent years, systematic evaluation of Chinese herbal medicines for cardiovascular disease have supported the notion that TCM formulations can be used as a complementary and alternative approach to primary and secondary prevention of cardiovascular disease [297]. Usually, TCM formulations are hot water extracts from a single plant or multiple-plant formulation. Some herbal medicines used for cardio-protection, such as Tongmanyangxinwan and Shenmai injection, have been shown to protect against cardiotoxicity in cancer treatment [298].

In a randomized controlled trial that began in 2016, 120 patients received twice-daily administration of *Pelargonium grandiflorum* granules or placebo to assess cardiac function by electrocardiography; left-ventricular diastolic function; and measuring physiological indicators such as brain natriuresis peptide, CK-MB, and troponin I to evaluate the cardioprotective effect and safety of *P. grandiflorum* in patients with early breast cancer receiving ANTs. That trial is ongoing [299]. One study suggested that *Hypericum hircinum* significantly reduced levels of lipid peroxidative (LP) value and marker enzymes of heart damage, increased GSH and SOD levels, reversed electrocardiography changes, and prevented heart-weight loss in a DOX-treated group [300]. Pretreatment with *Lycium barbarum* aqueous extract significantly prevented the loss of myogenic fibers and improved cardiac function in DOX-treated rats, as evidenced by lower mortality, restoration of antioxidant activity, normalization of serum levels of aspartate aminotransferase and CK, and improvement in arrhythmias and conduction abnormalities [301].

TCM formulations have been used to relieve the HF caused by chemotherapy. Cardiac edema improved after 2 months of treatment with modified Zhigancaotang (ZGCT) in a patient with refractory acute lymphoblastic leukemia [302]. Other botanical formulations with similar potential for clinical application have shown excellent therapeutic effects on the cardiotoxicity induced by cancer treatment in animal studies. Antarth is a polyherbal preparation. It was shown to inhibit the DOX-induced decrease in antioxidant status after pretreatment of Swiss albino mice but did not affect the anti-cancer activity of DOX. Its protective effect against cardiotoxicity was demonstrated by a significant reduction in DOX-induced increases in serum levels of GPT, GOT, CK-MB, and LDH [303]. The Tongmai Yangxin pill (TMYXP) has been shown to alleviate the cardiotoxicity induced by intraperitoneal injection of cisplatin. Oxidative stress is the main biological process of TMYXP, and Nrf2 and MAPK may be the key signaling pathway. In vivo studies have demonstrated that TMYXP improves resistance to oxidative stress and reduces apoptosis by modulating the Nrf2/HO-1 pathway and p38/MAPK pathway [298].

**Table 1 ijms-23-10617-t001:** Representative drugs to reduce cardiac toxicity caused by tumor treatment in published works.

Category	Compound/Herb Name	Source	Chemotherapy Drug	Experimental Model	Mechanism of Action	Ref.
Phenol acid	Salvianolic acid	*Salvia miltiorrhiza*	DOX	Male KM mice	Increased absorbance of oxygen radicals, decreased levels of creatine kinase and malondialdehyde.	[255]
	Curcumin	Turmeric	DOX	Kunming mice, primary neonatal rat cardiomyocytes	Regulated the 14-3-3γ/BAD/Bcl-2 pathway and mPTP.	[37]
	Resveratrol	Grapes and peanuts	DOX	Male Sprague-Dawley rats, rat embryonic cardiomyoblast-derived cells, H9c2 cells, male FBN rats	Regulated VEGF-B/Akt/GSK-3β signaling pathway; regulated mTOR, AMPK, and SIRT1 pathways.	[258]
Polysaccharides	*Ganoderma lucidum* polysaccharides	*Ganoderma lucidum*	DOX	H9c2, MCF-7, HepG2 cells	Regulated polyubiquitin modification, inhibited the NF-κB signaling pathway.	[262]
	*Astragalus* polysaccharide	*Astragalus membranaceus*	DOX	Neonatal rat ventricular myocytes, male C57BL/6 mice	Activated the PI3K/Akt pathway, inhibited the p38MAPK pathway, reduced levels of caspase 3 and caspase 9.	[263]
	*Ganoderma atrum* polysaccharide (PSG-1)	*Ganoderma atrum*	DOX	Kunming mice, primary myocardial cells	Improved antioxidant capacity, reduced cytochrome-c release, increased mitochondrial membrane potential, inhibited mPTP opening.	[264]
Flavonoids	Chrysin	Mushrooms, bee propolis, et al.	DOX	Male Sprague-Dawley rats	Regulated oxidative stress, as well as VEGF/PI3K/Akt, p53, p38, JNK, and NF-κB signaling pathways.	[271]
	Cardamonin	*Alpinia katsumadai*, *Ginkgo biloba*, and *Carya cathayensis* Sarg	DOX	Male C57BL/6 J mice, rat-myocardium-derived cardiomyoblast H9C2 and mouse cardiomyocyte HL-1	Regulated Nrf2, NF-κB, and caspase-3 pathways.	[272]
	Calycosin	*Radix astragali*	DOX	Rat cardiomyocyte line H9c2, male Kunming mice	Inhibited Bax; increased Bcl-2; activated PI3K/Akt pathway; reduced NLRP3, TXNIP, IL-1β, and caspase-1.	[273]
Saponin	Ginsenoside Rg3	*Ginseng*	DOX	Adult male Sprague-Dawley rats, cardiac microvascular endothelial cells from neonatal rats	Increased EF and FS, improved LV outflow, activated Nrf2-ARE and Akt pathways.	[280]
	Astragaloside IV	*Astragalus membranaceus*	DOX	C57BL/6 mice, NRCMs	Alleviated myocardial fibrosis and left-ventricular systolic dysfunction, inhibited NOX2 and NOX4.	[281]
	Saikosaponin D	*Radix bupleuri*	DOX	Male mice	Inhibited excessive oxidative stress via the p38MAPK signaling pathway.	[282]
Alkaloids	Neferine	*Nelumbo nucifera* Gaertn	DOX	Rat cardiomyoblasts, H9c2	Activated IGF-1R, PI3K/Akt/mTOR pathway, increased expression of SOD-1 and HO-1.	[294]
	Berberine	*Rhizoma coptidis*	DOX	Sprague-Dawley rats, rat cardiac H9c2 cell line	Improved ST, QRS complexes, and QT; regulated SIRT1-p66shc pathway.	[295]
	Cyclovirobuxine D	*Buxus microphylla*	DOX	C57BL mice injected (i.p.) with DOX	Ameliorated cardiac oxidative damage, prevented impairment of mitochondrial biogenesis.	[304]
Single herb	-	*Platycodon grandiflorum*	DOX	Clinical trial	Ongoing	[299]
	-	*Hypericum hircinum*	DOX	Wistar rats	Increased antioxidant defenses, reversed ECG changes, and prevented reductions in heart weight.	[300]
	-	*Lycium barbarum*	DOX	Male Sprague-Dawley rats	Prevented loss of myofibrils and improved heart function, normalized antioxidative activity.	[301]
TCM formulation	Modified ZGCT	*Radix Glycyrrhirae preparata*, *Radix Ginseng*, *Ramulus Cinnamomi*, *Colla Corii Asini*, *Radix Rehmanniae*, *Radix Ophiopogonis*, et al.	DOX	18-year-old male adolescent with refractory acute lymphoblastic leukemia	Chest radiograph showed great improvements in pulmonary edema and cardiomegaly.	[302]
	Antarth	*Boswellia serrata*, *Commiphora mukul*, *Withania somnifera*, *Smilax china*, *Tribulus terrestris*, *Curcuma longa*, et al.	DOX	12-week-old male Swiss albino mice	Inhibited DOX-induced decline in antioxidant status.	[303]
	Tongmai Yangxin pills	*Radix Rehmanniae*, *Radix Glycyrrhizae*, *Ramulus Cinnamomi*, *Fructus Schisandrae*, *Radix Ophiopogonis*, *Radix Polygoni multiflori preparata*, et al.	Cisplatin	Male Wistar rats	Improved anti-oxidative stress and reduced apoptosis through regulating Nrf2/HO-1 and p38MAPK pathways.	[298]

## 4. Advances in Reverse Cardiac Oncology

Advances in cardio-oncology have improved the adverse effects of anticancer therapy. However, the potential link between existing cardiovascular diseases and subsequent malignancies (reverse cardio-oncology) has been rarely studied. Patients with cardiovascular disease carry a higher risk of cancer at than that in the general population. Studies have shown that patients with HF have a 70% higher risk of developing cancer than patients without HF, and that cancer risk increases over time [305]. The Randomized Evaluation Long-Term Anticoagulant Therapy Study (RE-LY) showed that more than one-third of deaths in people with atrial fibrillation were due to non-cardiovascular causes, and malignancies were the leading cause those deaths [12]. Because patients with cardiovascular disease have a higher risk of cancer, reverse cardio-oncology is beginning to gain attention. Two major underlying mechanisms are involved in heart disease-induced tumor growth: (i) secreted factors; (ii) reprogramming of immune cells [306] (Figure 6).

Meijers et al. [16] demonstrated, for the first time, the relationship between HF and development of intestinal tumors through animal experiments. They ligated the left anterior descending coronary artery in APC^min^ mice to induce massive myocardial infarction, which caused HF. Many polyps were found in the intestine of mice with HF. To study further how HF promoted tumor formation, they transplanted the (extra) infarcted heart into the cervical region of recipient APC^min^ mice to exclude the effect of hemodynamic impairment: HF continued to promote tumor development. Subsequent screening identified the circulating protein serpinA3 as promoting the development of colorectal cancer, and this finding was validated by cell experiments. Koelwyn et al. [307] found that myocardial infarction could accelerate tumor growth in an orthotopic model of breast cancer in mice. Selective depletion of CCR2^+^ monocytes resulted in reduced tumor growth in mice suffering from HF. Further analyses of the intratumoral immune landscape revealed that the increased tumor growth was accompanied by an epigenetically induced increase in monocytic myeloid-derived suppressor cells in tumor tissue. Bone marrow transplantation demonstrated that epigenetic changes in the bone marrow after myocardial infarction persisted after transplantation into naive mice, where they drove myelopoiesis and accelerated tumor growth. The authors undertook surgery for myocardial infarction in a model of spontaneous breast cancer (MMTV (mouse mammary tumor virus)—PyMT (polyoma middle T-antigen)) and came to similar conclusions. Those studies provide conclusive evidence that HF accelerates tumor growth.

Avraham et al. [308] investigated whether early cardiac remodeling promotes tumor growth in the absence of HF. They used the transverse aortic constriction (TAC) model combined with two cancer models, an orthotopic breast cancer model (PyMT) and a syngeneic lung cancer model (Lewis lung carcinoma cells), finding that early cardiac remodeling accelerated tumor development in both models. Tail vein injection of tumor cells revealed that TAC promoted the metastasis of these two types of cancer. Under the exclusion of immune factors, they confirmed that the serum of mice undergoing TAC could promote tumor cell growth through in vitro experiments and demonstrated that connective tissue growth factor and periostin had a decisive role in this process. Similarly, cardiac hypertrophy caused by drug intervention [309], as well as cardiac hypertrophy caused by altered gene expression [310], has also been shown to accelerate tumor growth and to lead to colonization of metastases through changes in secreted proteins. Those studies add to the evidence that heart disease accelerates tumor growth, and that cardiac remodeling can promote tumor metastasis. However, the effect of HF on tumor growth is not universal. Shi et al. [311] carried out orthotopic transplantation of kidney tumors in an HF model in mice. They demonstrated that HF did not accelerate the progression of kidney cancer by comparing emitted bioluminescent signals, tumor weight, kidney volume, and distant metastases. Hence, although there is evidence that certain heart diseases promote tumor development, these effects may depend on the specific type/subtypes of cancer and heart disease, and broad generalizations cannot be made.

Circulating miRs may become new targets for reverse cardio-oncology studies. Katoh and colleagues [312] reported that miRs involved in heart disease, vascular disease, and cancer were designated as “cardio-miRs”, “angio-miRs”, and “onco-miRs”, respectively. However, there was an intersection among these separately defined miRs. For example, miR-125b expression is upregulated in cardiac hypertrophy and HF, and miR-125b in human cancers promotes the proliferation and survival of tumor cells through repression of BCL2 antagonist/killer 1 (BAK1), p14^ARF^ (ARF-tumor-suppressor protein), suppression of tumorigenicity 18 (ST18), and tumor protein p53 (TP53). In addition, miR-24 expression is upregulated in the chronic phase after myocardial infarction and promotes the hypertrophic growth of cardiomyocytes [313]. miR-24 achieves its oncogenic role by negatively regulating the inhibitor of growth 5 (ING5) expression in breast cancer [314]. Furthermore, a recent review proposed a potential link between miR-208a and reverse cardio-oncology. On the one hand, the increase in miR-208a expression promotes ROS production, and the change in redox balance caused by the increase in ROS promotes the occurrence and development of tumors and is an important mechanism for the occurrence of cardiovascular diseases. Hence, this phenomenon may lead to crosstalk between disease mechanisms. On the other hand, acute myocardial infarction increases expression of cardiac-enriched miRs (including miR208a) in circulating exosomes, and the release of such circulating exosomes could affect bone marrow stem cells. Likewise, the authors propose that this circulating miR can promote the proliferation and invasion of tumor cells in the form of exosomes. These works demonstrated that miRs may be important regulators of various cardiovascular diseases and pathological processes in cancer. In-depth understanding of the role of miRs in reverse cardio-oncology is expected to “kill two birds with one stone” [315].

## 5. Conclusions and Outlook

Rapid advances in cardio-oncology have expanded our understanding of the molecular mechanisms underlying the cardiovascular toxicity associated with cancer therapy. However, several uncertainties remain regarding the inducing mechanism of cardiotoxicity. From the perspective of autophagy, whether the cardiomyocyte or mitochondrial dysfunction caused by cancer treatment is due to the inhibition of autophagy or activation of autophagy is unclear and lacks a key measurement standard. In addition, cancer immunotherapy, as a new treatment method, has been reported to elicit cardiotoxicity. However, the specific pathological mechanism is incompletely understood, and the cardiotoxicity caused may be underestimated. The cardiotoxicity mechanisms of antitumor therapy are complex and diverse, but no studies have compared these mechanisms, and which mechanism contributes more to cardiotoxicity in different treatment methods remains to be studied. The current hope is that proven cardiotoxicity-modifying drugs can be manipulated and adjusted to the patient’s physical status, thereby enabling “personalized therapy”. The rational use of TCM formulations is also important. As cardiology and oncology have become research “hotspots”, the development of reverse cardio-oncology is inevitable. Experimental evidence on whether cardiovascular disease promotes tumor progression needs to be supplemented. According to the available evidence, heart disease can promote certain types of tumor processes. Hence, should we strengthen the detection and prevention of certain specific tumors in patients with heart disease? The task of solving these problems remains daunting.

## Figures and Tables

**Figure 1 ijms-23-10617-f001:**
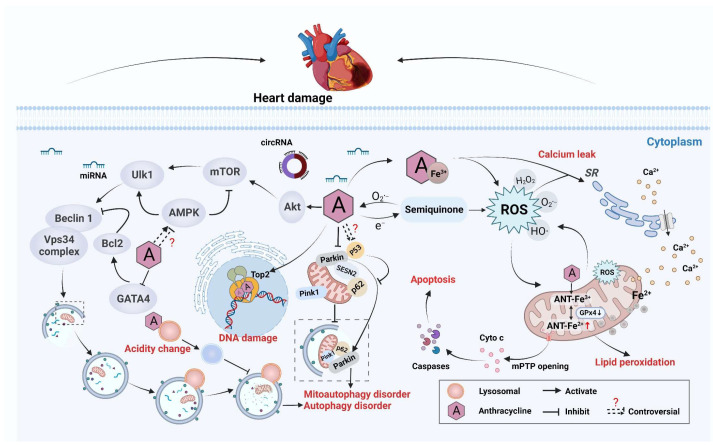
ANTs cause cardiotoxicity through multiple mechanisms. ANTs induce ROS production: a large number of ROS causes mitochondrial swelling, opens the mPTP, and activates endogenous apoptotic pathways. ANTs cause mitochondrial lipid peroxidation by downregulating GPx4 expression. ANTs cause dysregulation of macroautophagy and mitophagy and affect autophagy-lysosomal acidity. ANTs targeting Top2 leads to DNA damage. ANTs induce Ca^2+^ leakage in the sarcoplasmic reticulum. ANTs induce changes in ncRNA expression. ROS, reactive oxygen species;mTOR, mammalian target of rapamycin; Bcl-2, B-cell lymphoma 2; Ulk1, unc-51-like autophagy activating kinase 1; SR, sarcoplasmic reticulum; Top2, topoisomerase 2. Created with BioRender.com.

**Figure 2 ijms-23-10617-f002:**
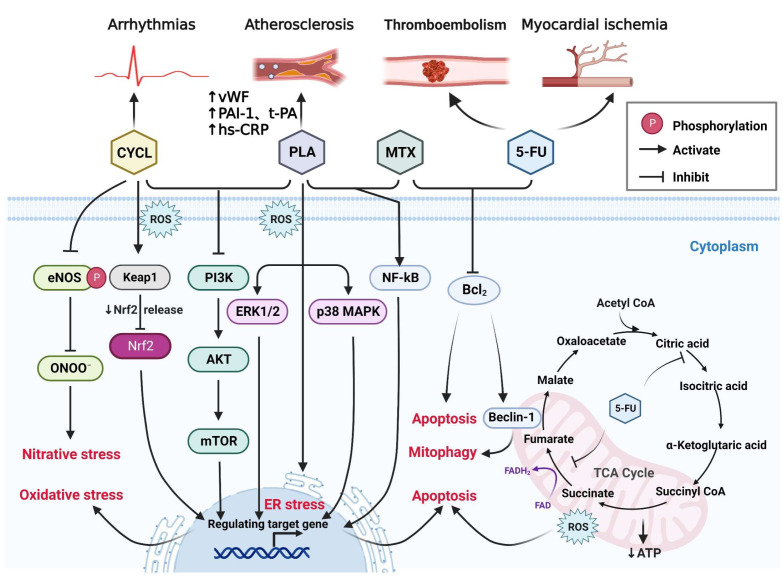
Mechanisms of non-anthracycline-based chemotherapeutic cardiotoxicity. The mechanisms of cisplatin toxicity to the heart include oxidative stress, inflammation, endoplasmic reticulum stress, and activation of ERK1/2 and p38 MAPK pathways. Cyclophosphamide causes myocardial injury mainly by inducing oxidative stress, nitrative stress, and inhibiting the PI3K/AKT/mTOR pathway. MTX activates proinflammatory pathways, inhibits Bcl2 expression, and induces cardiomyocyte apoptosis. 5-FU activates mitophagy and blocks the TCA cycle, thereby causing mitochondrial dysfunction. CYCL, cyclophosphamide; PLA, platinum; vWF, von Willebrand factor; t-PA, tissue plasminogen activator; hs-CRP, high-sensitivity C-reactive protein, PAI-1, plasminogen activator inhibitor; NF-κB, nuclear factor kappa B; eNOS, endothelial nitric oxide synthase; Keap1, Kelch-like ECH-associated protein 1; Nrf2, nuclear factor erythroid 2-related factor 2. Created with BioRender.com.

**Figure 3 ijms-23-10617-f003:**
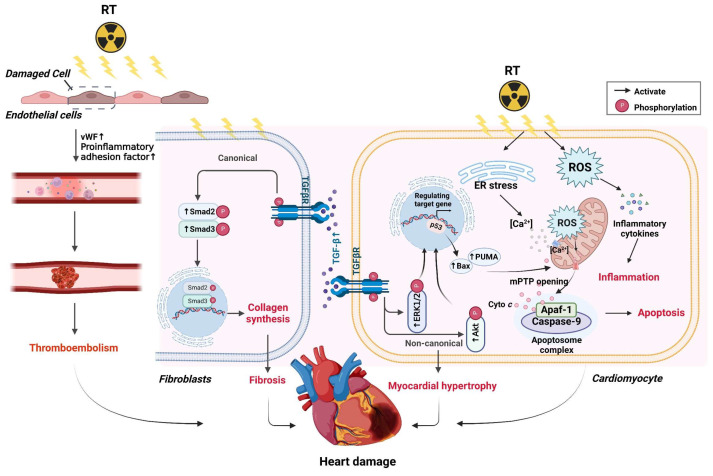
Mechanisms of radiotherapy-induced cardiotoxicity. Radiotherapy stimulates Ca^2+^ release from the ER, causing Ca^2+^ overload in mitochondria. Radiation-produced ROS leads to mitochondrial dysfunction and activation of endogenous apoptotic pathways. ROS can induce an inflammatory response. Radiation causes increased expression of TGF/β, which induces myocardial hypertrophy by targeting ERK1/2 and AKT signaling pathways. Radiotherapy increases the risk of thrombosis by upregulating expression of vWF and proinflammatory adhesion factors, as well as causing endothelial damage. RT, radiotherapy; Bax2, bcl-2-associated X protein; Apaf-1, apoptotic protease activating factor-1; PUMA, polyunsaturated fatty acid; Cyto c, cytochrome c. Created with BioRender.com.

**Figure 4 ijms-23-10617-f004:**
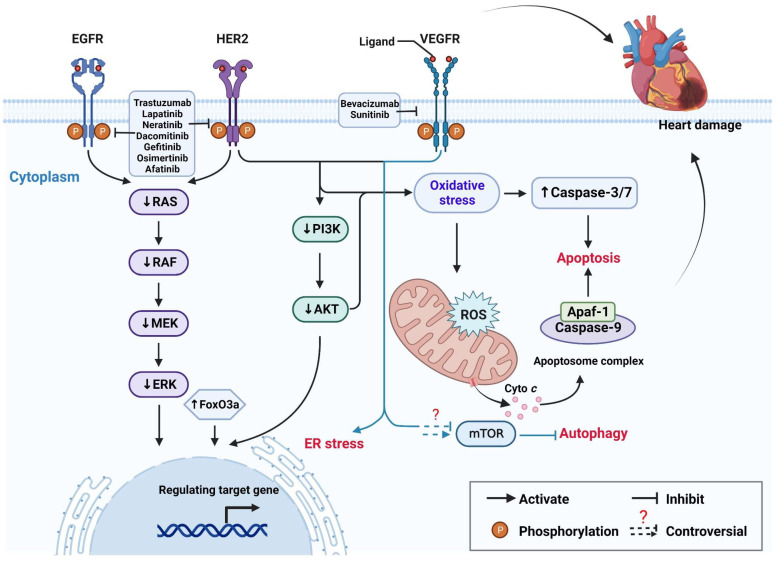
Mechanisms of cardiotoxicity induced by monoclonal antibodies/small molecule tyrosine kinase inhibitors. Inhibition of intracellular signaling by interference with downstream signaling pathways induces oxidative stress, ER stress, and apoptosis, ultimately causing cardiac injury. EGFR, epidermal growth factor receptor; HER2, human epidermal growth factor receptor 2; VEGFR, vascular endothelial growth factor receptor. Created with BioRender.com.

**Figure 5 ijms-23-10617-f005:**
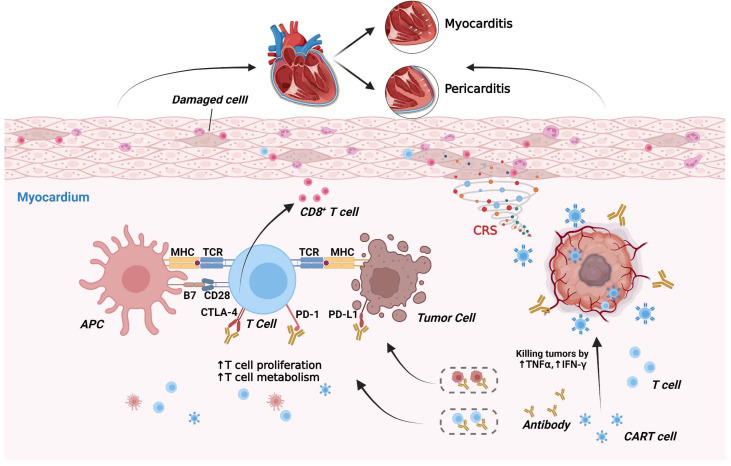
Mechanisms of immunotherapy-induced cardiotoxicity. Inhibition of CELA-4 expression leads to the proliferation and infiltration of CD8^+^ T cells in the heart. Chimeric antigen receptor T-cell therapy drives CRS to an inflammatory response. APC, antigen-presenting cells; MHC, major histocompatibility complex; TCR, T-cell receptor; CTLA-4, cytotoxic T-lymphocyte-associated antigen-4; PD-1, programmed cell death protein 1; PD-L1, programmed cell death 1 ligand 1; CRS, cytokine release syndrome. Created with BioRender.com.

**Figure 6 ijms-23-10617-f006:**
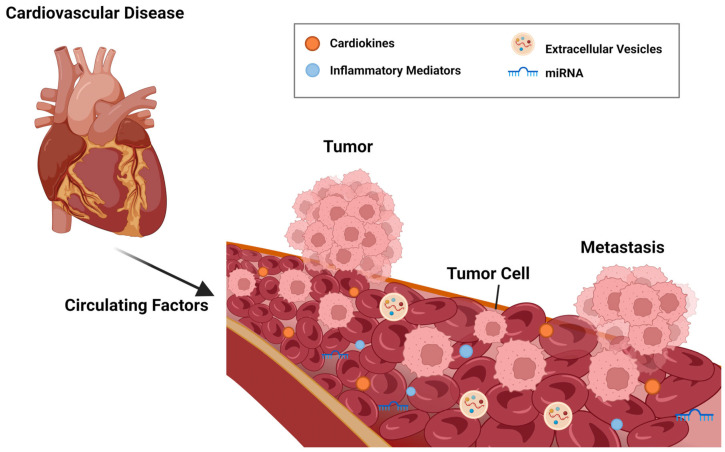
Mechanisms of reverse cardiac oncology (schematic). Secreted factors from the damaged heart promote the growth and metastasis of tumor cells through the circulation. Created with BioRender.com.

## Data Availability

Not applicable.

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
