# Peer review of "Cardio-Oncology: Mechanisms, Drug Combinations, and Reverse Cardio-Oncology"

_ijms, 2022, doi:10.3390/ijms231810617_

Round 1
Reviewer 1 Report
The paper is interesting and has the potential to publish in the considered journal after the authors will address the following major comments:
- Nomenclature must be added to the paper. In order to allow readers who are not from the field to read and understand the current research.
- The general impression is that the article presents data and information that are known in the biological/medical/oncological field and there is very little research presented in the article.
- And I wonder if this is a review article or a research article with relevant results
- The following paper can be relevant:
1 Han, X., Zhou, Y. & Liu, W. Precision cardio-oncology: understanding the cardiotoxicity of cancer therapy. npj Precision Onc 1, 31 (2017). https://doi.org/10.1038/s41698-017-0034-x
2 Analysis of a breast cancer mathematical model by a new method to find an optimal protocol for HER2-positive cancer OP Nave, M Elbaz, S Bunimovich-Mendrazitsky Biosystems 197, 104191
-
Author Response
Comments: The paper is interesting and has the potential to publish in the considered journal after the authors will address the following major comments.
Point1:Nomenclature must be added to the paper. In order to allow readers who are not from the field to read and understand the current research.
Response 1: We thank the reviewer for this good suggestion, and we have added a table to summarize the nomenclatures and abbreviations at the end of our manuscript so that readers can read and understand the current research easily.
Point 2:The general impression is that the article presents data and information that are known in the biological/medical/oncological field and there is very little research presented in the article.
And I wonder if this is a review article or a research article with relevant results.
Response 2: The type of our manuscript is a review article, which summarizes and discusses studies in both cardio-oncology and reverse cardio-oncology.
Our review article highlights three aspects of relevant studies:
- Molecular mechanisms underlying the cardiotoxicity caused by cancer treatments.
- Clinical drugs and preclinical chemicals that could alleviate the cardiotoxicity caused by cancer treatments.
- The progresses in reverse cardio-oncology—an emerging field of research.
Point 3:The following paper can be relevant:
1 Han, X., Zhou, Y. & Liu, W. Precision cardio-oncology: understanding the cardiotoxicity of cancer therapy. npj Precision Onc 1, 31 (2017). https://doi.org/10.1038/s41698-017-0034-x.
2 Analysis of a breast cancer mathematical model by a new method to find an optimal protocol for HER2-positive cancer OP Nave, M Elbaz, S Bunimovich-Mendrazitsky Biosystems 197, 104191.
Response 3: We thank the reviewer for these nice recommendations, and we have discussed and referenced these articles in the revised manuscript. (Reference 18, 197)

Reviewer 2 Report
This is a well organized and comprehenive piece of work, which has merit for publication.
I believe that in the targeted therapy section, the work could take advantage by making reference to the PMID: 29455670, as well as to the PMID: 31627329 in th radiotherapy paragraph.
Overall is a nice and interesting work
Author Response
Comments: This is a well organized and comprehenive piece of work, which has merit for publication.
Point 1:I believe that in the targeted therapy section, the work could take advantage by making reference to the PMID: 29455670, as well as to the PMID: 31627329 in th radiotherapy paragraph.
Overall is a nice and interesting work.
Response 1: We thank the reviewer for his favorite comments on our manuscript. We also appreciate the wonderful recommendations from the reviewer, and we have discussed and referenced these articles in the revised manuscript. (Reference 215, 214)

Round 2
Reviewer 1 Report
The authors revised the paper partially according to my comment except for form the papers that I recommended adding to the list of references.
The paper needs to edit by English native language